# A landscape of gene regulation in the parasitic amoebozoa *Entamoeba* spp

**Edgardo Galán-Vásquez**[1]*, **María del Consuelo Gómez-García**[2], **Ernesto Pérez-Rueda**[3]*

1 Departamento de Ingeniería de Sistemas Computacionales y Automatización, Instituto de Investigaciones en Matemáticas Aplicadas y en Sistemas, Universidad Nacional Autónoma de México, Ciudad Universitaria, Ciudad de México, México, 2 Laboratorio de Biomedicina Molecular, Escuela Nacional de Medicina y Homeopatía, Instituto Politécnico Nacional, Ciudad de México, México, 3 Unidad Académica Yucatán, Instituto de Investigaciones en Matemáticas Aplicadas y en Sistemas, Universidad Nacional Autónoma de México, Mérida, Yucatán, México

* edgardo.galan@iimas.unam.mx (EG-V); ernesto.perez@iimas.unam.mx (EP-R)

**Data Availability Statement:** All relevant data are within the manuscript and its Supporting Information files.

**Funding:** This work was supported by the Dirección General de Asuntos del Personal

## Abstract

*Entamoeba* are amoeboid extracellular parasites that represent an important group of organisms for which the regulatory networks must be examined to better understand how genes and functional processes are interrelated. In this work, we inferred the gene regulatory networks (GRNs) in four *Entamoeba* species, *E. histolytica*, *E. dispar*, *E. nuttalli*, and *E. invadens*, and the GRN topological properties and the corresponding biological functions were evaluated. From these analyses, we determined that transcription factors (TFs) of *E. histolytica*, *E. dispar*, and *E. nuttalli* are associated mainly with the LIM family, while the TFs in *E. invadens* are associated with the RRM_1 family. In addition, we identified that EHI_044890 regulates 121 genes in *E. histolytica*, EDI_297980 regulates 284 genes in *E. dispar*, ENU1_120230 regulates 195 genes in *E. nuttalli*, and EIN_249270 regulates 257 genes in *E. invadens*. Finally, we identified that three types of processes, Macromolecule metabolic process, Cellular macromolecule metabolic process, and Cellular nitrogen compound metabolic process, are the main biological processes for each network. The results described in this work can be used as a basis for the study of gene regulation in these organisms.

## Introduction

The passage of information from DNA to RNA (transcription) is a fundamental mechanism for all organisms [1]. The mechanism for transcription involves a large number of molecules (proteins, enzymes, and DNA sequences, among others) that together orchestrate and carry out the expression of genes in a highly precise, spatially and temporally controlled manner to meet the needs of the cell. Transcription factors (TFs) are essential proteins in this event and are part of the cell's ability to have differential and temporal expressions, increases or decreases in the amounts of transcripts, etc. They do this by interacting with cis-consensus DNA sequences present in gene promoters and with general TFs [2].

Académico-Universidad Nacional Autónoma de México (IA201221 and IN-209620), as well as the Secretaría de Investigación y Posgrado, Instituto Politécnico Nacional [Grant SIP20211325] and CONACYT (320012). The funders had no role in study design, data collection and analysis, decision to publish, or preparation of the manuscript.

**Competing interests:** The authors have declared that no competing interests exist.

TFs may have different types of domains through which they bind to DNA, interact with other proteins to regulate activation, and locate spatially within the cell, and they have motifs through which they undergo different types of posttranslational modifications, such as phosphorylation, acetylation, methylation, SUMOylation, and ubiquitination [3], which are crucial for TF function. Numerous studies on TFs have demonstrated how conserved they are and their specificity for binding to DNA in various species, such as mice [4]. However, protozoa are a group for which very little is known about transcription and TFs.

Particularly in the *Entamoebidae* family, parasitic species have been described, including *Entamoeba histolytica*, the causal agent of intestinal amebiasis and amoebic liver abscess in humans, which causes 100,000 annual deaths [5–9]. *E. nuttalli* infects macaques and different species of monkeys and causes intestinal amebiasis, liver abscesses, and even death. This species is responsible for serious health problems in zoos and in various regions of the planet, such as nature reserves [10–12]. *E. invadens* is a parasite of reptiles (ophidians, saurians, and chelonians) and causes gastrointestinal damage from mild to severe and is the only species of the genus *Entamoeba* that encysts *in vitro* [13]. Finally, *E. dispar* is considered a nonpathogenic species that lives as a commensal in humans; first described in 1993 for Diamond and Clark [14] was posteriorly identified as able to produce liver and intestinal lesions that were occasionally indistinguishable from those produced by *E. histolytica* [15].

These four species of *Entamoeba* are amoeboid extracellular parasites that move via the emission of pseudopods and lack mitochondria; this is why they are located in the early phase of eukaryotic evolution [16]. Only in *E. histolytica* and more recently in *E. invadens* has the presence of mitosomes been described [17,18]. These organisms present two phases in their life cycle: the cyst, which is the infective form, and the trophozoite, which is the invasive form [16,19].

These four species of *Entamoeba* are undoubtedly of great importance in the parasitology not only of humans, but also of other organisms. Therefore, the genomes of these four species have already been sequenced. *E. histolytica* has a genome rich in AT (75%) with a size of 20,800,560 bp and 8,333 genes [20]; *E. dispar* has a genome similar to that of *E. histolytica*, with 8,749 genes and a composition of 76.5% AT and size of 22,955,291 bp. *E. nuttalli* has a genome that contains 74.9% AT, a size of 14,399,953 bp, and 6,193 genes, and of the four mentioned species it has the smallest genome [21]. *E. invadens* has the largest genome of these four species, with 40,888,805 bp, and AT content of 70%, and 11,549 genes [22].

A gene regulatory network (GRN) is a directed graph in which interaction edges connect TFs to target genes (TGs) [23]. This type of network greatly helps in understanding the links between genes and the products they encode, which is a crucial and difficult step in experimental and computational biology [24]. Only a few GRNs have been reconstructed in model organisms [24–27]. As a result, homology-based techniques are frequently used to research GRNs in species that are less well-known [28–31].

In this work, we inferred the GRNs of the four main *Entamoeba* species using a criterion of TF-TG orthology relationships from reference GRNs experimentally described. The reconstructed GRNs were posteriorly analyzed in terms of topology. We consider that the GRN inferences for these strains open the opportunity to explore organisms of public health importance.

## Material and methods

The network inference process steps were described in the schematic workflow (Fig 1). Details on each step, including input and output data, are described in follow.

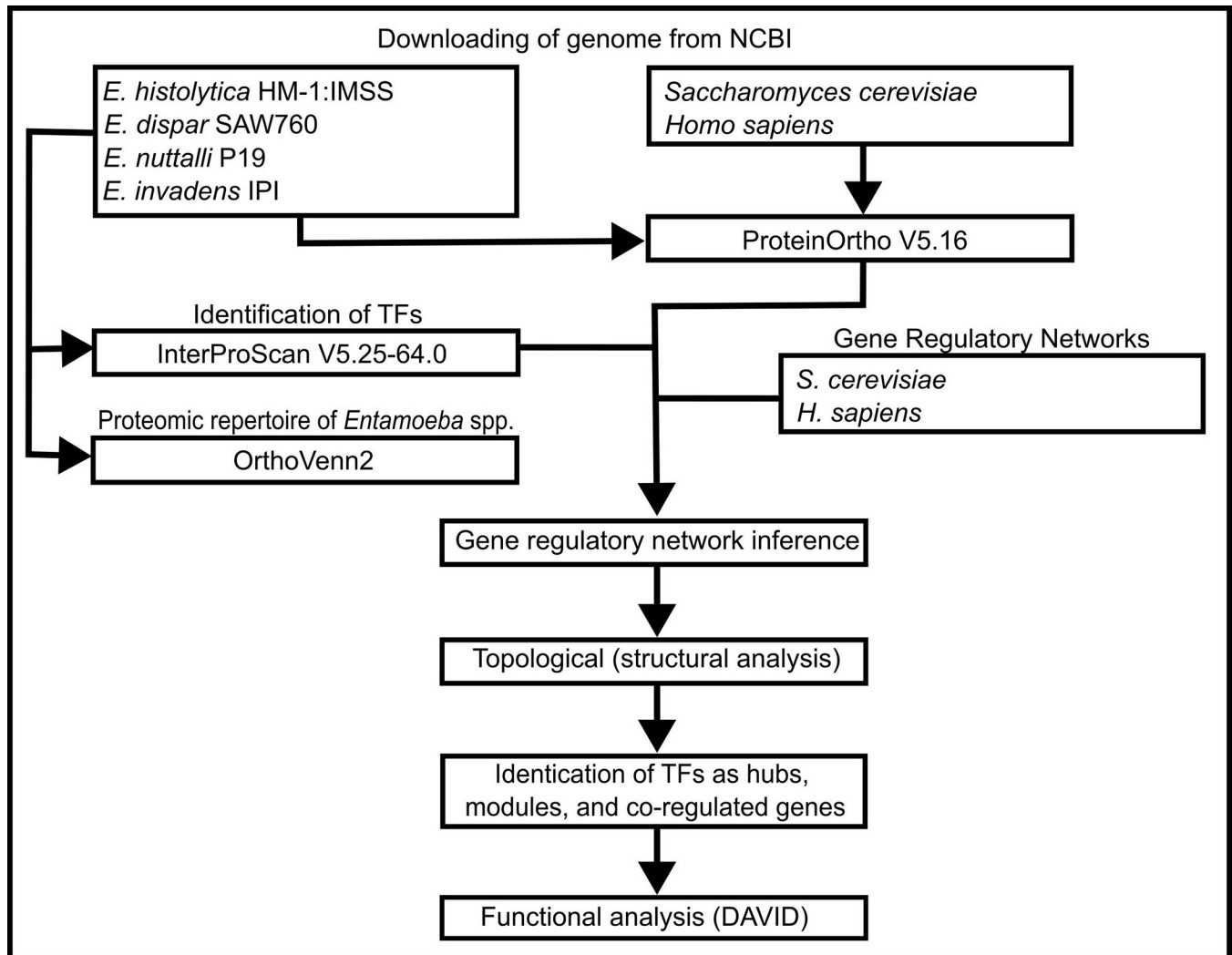

**Fig 1. Schematic workflow of the network inference procedure steps.** Four Entamoeba genomes were downloaded from NCBI and compared with Orthovenn, to infer the common set of orthologous proteins. Interproscan was used to assign protein domains (Pfam and Supfam). To infer the GRN, two organismal models were used, S.cerevisiae and H. sapiens, and the ProteinOrtho was considered. The GRNs inferred were evaluated in terms of their topological properties, hubs, and functional descriptions.

## Genomes analyzed

Four genomes of *Entamoeba* spp. were downloaded from the NCBI server: *E. histolytica* HM-1:IMSS (GCF_000208925.1), *E. dispar* SAW760 (GCF_000209125.1), *E. nuttalli* P19 (GCF_000257125.1), and *E. invadens* IP1 (GCF_000330505.1). Additionally, the genomes of *Saccharomyces cerevisiae* (GCF_000146045.2), and *Homo sapiens* (GCF_000001405.39) were downloaded to be used for the GRN inferences.

## Proteomic repertoire in *Entamoeba* spp.

OrthoVenn2 was used to identify orthologous clusters in the four proteomes, and to perform a functional enrichment analysis for each cluster, we used an E-value of 0.01 as cutoff for all-to-all protein similarity comparisons, and an inflation value of 1.5 for the orthologous clustering employing the Markov Cluster algorithm. The enrichment analysis was considered significant with a P-value less than 0.05 [32].

## Identification of TFs

To assess the TF diversity, protein sequences of whole proteomes were used to identify DNA-binding domains (DBDs) associated with regulatory proteins. To do this, InterProScan (v5.25–64.0) [33] was used to map InterPro families and DBDs, using default parameters. Afterwards, 162 Pfam IDs obtained from the TF database and by literature lookup were compiled and identified in the associated predictions (S1 Table).

## Reconstruction of GRNs

Two organisms were considered templates for the inferences of the GRNs of *Entamoeba*. The GRN of *S. cerevisiae* was obtained from the YEASTRACT database and is composed of 6,709 nodes and 179,601 edges [34]. The GRN of *H. sapiens* with 2,862 nodes and 8,427 edges was obtained from the TRRUST database v2 [35].

To identify orthologous proteins between the four *Entamoeba* proteomes and the proteomes of *S. cerevisiae* and *H. sapiens*, were used to identify the orthologs using the program ProteinOrtho (V5.16) [36], with the following parameters: E-value of 0.01, a sequence coverage ≥ 50%, and minimal percent identity of best Blast hits of 30%.

To infer the GRNs, we map their interactions considering the following criteria: If the orthologs of the TF and its TG of the model organism (*S. cerevisiae* and / or *H. sapiens*), were found in a new genome, the interaction was assigned using guilt by association. Then, each network was integrated using all the ortholog assignments with the two six reference GRNs. All the network interactions can be inferred by running the scripts, provided as supplementary data 1.

## Network structural analysis

To determine the topologies of the reconstructed networks, the following metrics were calculated: the number of edge incidents with other nodes, *i.e.*, the Node degree ($K$). In GRNs, input degree ($Kin$) is the number of arrows that enter anode, which corresponds to the TFs that affect a TG, and output degree ($Kout$) is the number of arrows that leave a node, which corresponds to the number of TGs by which a TF is regulated [37,38].

A clustering coefficient measures how connected a node's neighbors are to one another. It is calculated as, "the number of edges connecting a node's neighbors divided by the total number of possible edges between the node neighbors" [39]. The connectivity in an undirected network is the link between two nodes, and this link can be via a direct or indirect edge through intermediate connections. In this context, a connected component is a set of nodes that are linked to each other node by a path, and the component with the most proportions of nodes is called a giant component [38].

Some metrics are proposed to identify the relevance of nodes in a network. Hubs, which are defined as the most connected nodes with other nodes, confer the global structure of the network. Centrality ($C$), which measures the contribution or importance of nodes, sets node $u$ as more important than another node $v$ if $C(u) > C(v)$. The most relevant centrality metrics are: degree, closeness, betweenness, and eigenvector centrality, which assigns every $v \in V$ of a given graph $G$ a value $C(v) \in R$ [38].

## Functional annotation analysis

To determine the function enriched in each network, we used the Database for Annotation, Visualization and Integrated Discovery (DAVID 6.81), a gene functional classification system that integrates a set of functional annotation tools [40]. Each list of genes from the networks

was used to perform an enrichment analysis in Gene Ontology terms, and a statistical significance at a P-value of < 0.05 was set.

## Results and discussion

### Protein similarities of *Entamoeba* spp.

In order to evaluate how similar, the proteomic repertoire of *Entamoeba* species are, we analyzed the shared orthologous proteins between *E. histolytica*, *E. nuttalli*, *E. invadens*, and *E. dispar* genomes and displayed them in OrthoVenn2 [32]. The OrthoVenn revealed 4,285 clusters of 18,545 orthologous proteins that are shared by all species, accounting for 55.86% of the *E. histolytica* proteome, 51.93% of the *E. dispar* proteome, 70.73% of the *E. nuttalli* proteome, and 41.95% of the *E. invadens* proteome (Fig 2). The main functions associated with these proteins correspond to metabolic processes (GO:0008152), cellular processes (GO:0009987), and macromolecule metabolic processes (GO:0043170). The second longest group of 1,154 clusters includes only proteins of *E. nuttalli* (1,157 proteins), *E. histolytica* (1,205 proteins), and *E. dispar* (1,222 proteins) (Fig 2). These results show that the four species share ~50% of their proteins, which makes sense, as they are organisms of the same genus and all of them can infect a host. However, it is intriguing that *E. dispar*, shares 51.93% of its proteome with *E. histolytica*, a species capable of colonizing the human intestine and even inducing the development of amoebic liver abscesses, the more serious form of disease; this suggests that other factors such as host or environmental factors, in addition to some genetic factors, may define the pathogenicity of these two species. Likewise, the observation that *E. nuttalli*, a species whose host is the macaque, shares a very high percentage of proteins with the other three species is striking, particularly because it has been shown that *E. nuttalli* is the closest species to *E. histolytica* [8,16].

On the other hand, we identified singletons, which are proteins not grouped in any cluster. In this context, *E. histolytica* contains 626 singletons that are mainly related to organic substance metabolic process, primary metabolic process, and cellular metabolic process; *E. dispar*

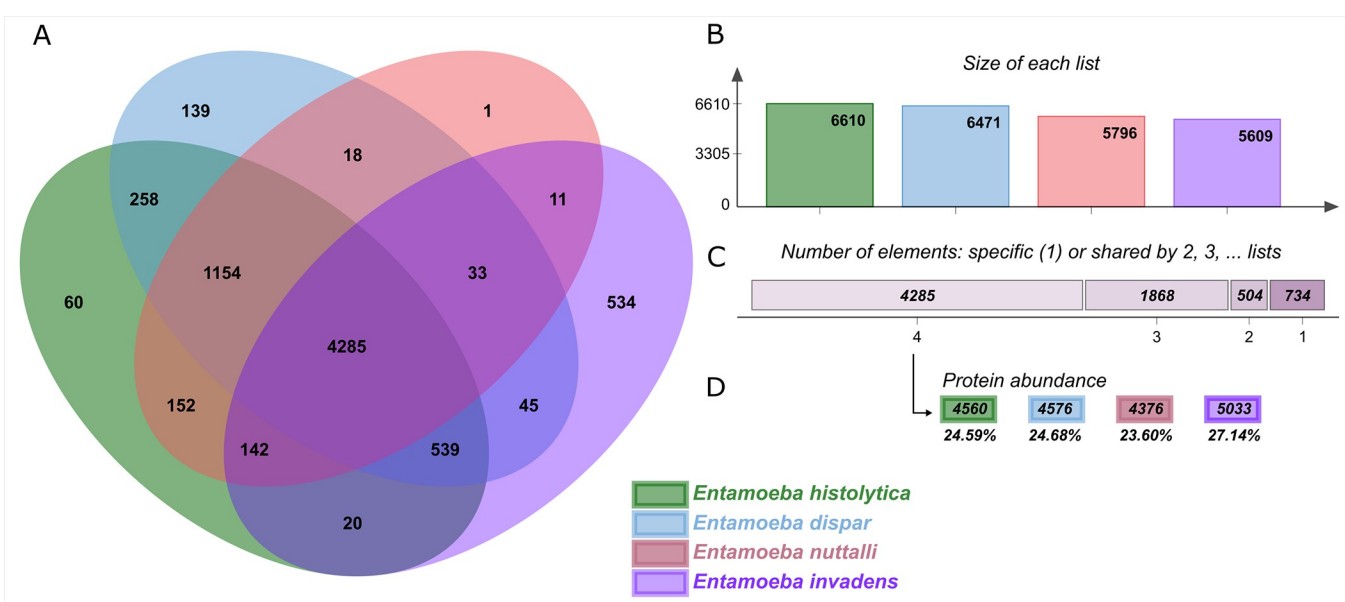

**Fig 2. Orthologous proteins shared between Entamoeba strains.** A) Orthologous clusters of whole proteomes. B) The bar plot graph shows the number of orthologous clusters by organism. C) The plot indicates the number of clusters that are organism specific or shared by 2, 3, or 4 organisms. D) For the 4,285 clusters shared by 4 organisms, the protein abundance levels are shown for each organism.

contains 1,137 singletons mainly related to primary metabolic process, organic substance metabolic process, and cellular metabolic process; *E. nuttalli* contains 286 singletons mainly related to biosynthetic process and single-organism signaling. Finally, *E. invadens* contains 2,927 singletons mainly related to single-organism cellular process, cellular response to stimulus, regulation of cell process, and single-organism signaling. These results show that *E. invadens* presents more single proteins than *E. dispar*, *E. histolytica*, and *E. nuttalli*. This may be because *E. invadens* has a much larger genome that is almost twice the size of the genomes of the other three species. Therefore, the genetic information contained in *E. invadens* is probably necessary to infect different species of reptiles, such as turtles, lizards, and snakes [41]. Thus, the processes and mechanisms in which these proteins are involved (cellular process, cellular response stimulus, and regulation of cell process) are different from those of *E. histolytica* and *E. dispar* and to a lesser extent those of *E. nuttalli*. A similar result was previously described for *E. histolytica*, *E. dispar*, *E. invadens*, and *E. moshkowskii* [16]. Therefore, amoebic genetic diversity may vary depending on host species.

## Identification of transcription factors

A TF repertoire consists of a set of proteins that regulate gene expression in the cell. Based on the PFAM assignments from InterProScan, we identified a set of 242 TFs in *E. histolytica*, 297 TFs in *E. invadens*, 210 TFs in *E. nuttalli*, and 245 TFs in *E. dispar*, representing 2.9%, 2.57%, 3.39%, and 2.8% of each proteome, respectively. The number of TFs identified in each species of *Entamoeba* seems not to be so different between them, nor to be related to the size of their genomes. However, the number of TFs obtained for each species is within the range of TFs stipulated for other organisms, as it is estimated that TFs constitute between 0.5 and 8% of the genes contained in the genomes of eukaryotic organisms [42].

Interestingly, the TFs predicted in *E. histolytica* are distributed among 11 families, whereas the TFs predicted in *E. invadens* are distributed among 87 families, 103 families in *E. nuttalli*, and 99 families in *E. dispar*. The most abundant family in *E. histolytica*, *E. dispar*, and *E. nuttalli* is the LIM domain (PF004112) (Fig 3), which is a protein structural domain containing two Zn2+ fingers separated by a 2-amino-acid hydrophobic linker. The LIM domain can bind a wide variety of protein targets and is widely distributed among plants, fungi, protozoa, and animals [43,44]. However, to date, only in the species *E. histolytica* has a protein with this domain, called EhLimA, been identified; EhLimA is associated with the actin of the parasite cytoskeleton and membrane [44]. Several possible LIM proteins have been identified in the *E. histolytica* genome, some of which could be relevant proteins in transcription for this *Entamoeba* species [45]. On the other hand, the TFs of *E. invadens* are associated with the RRM_1 family (PF00076) (Fig 3), which is a putative RNA-binding domain of approximately 90 amino acids and is known to bind single-stranded RNAs [46]; it is found abundantly in all life kingdoms [47,48]. Nevertheless, no protein with this domain has been characterized in any *Entamoeba* species to date. Interestingly, the proteins that contain this domain participate in the preprocessing of mRNAs, alternative splicing, stability, edition, and export of mRNAs, and thus they are fundamental in the biological processes of the organism [49]. This contrast in the number of families in which the TFs of these four *Entamoeba* species are distributed may be due to the characteristics of each species, that is, their forms and lifestyles, structures, and life cycles between other factors that have allowed them to specialize according to their needs. For example, *E. invadens* has the ability to infect different types of reptiles (snakes, turtles, and lizards), organisms in which osmotic changes or abrupt carbon source depletion in the intestine can be common, whereas in the human intestine such abrupt changes do not occur [50]. Thus, the type of TF families other *Entamoeba* species require may be different, even having species-

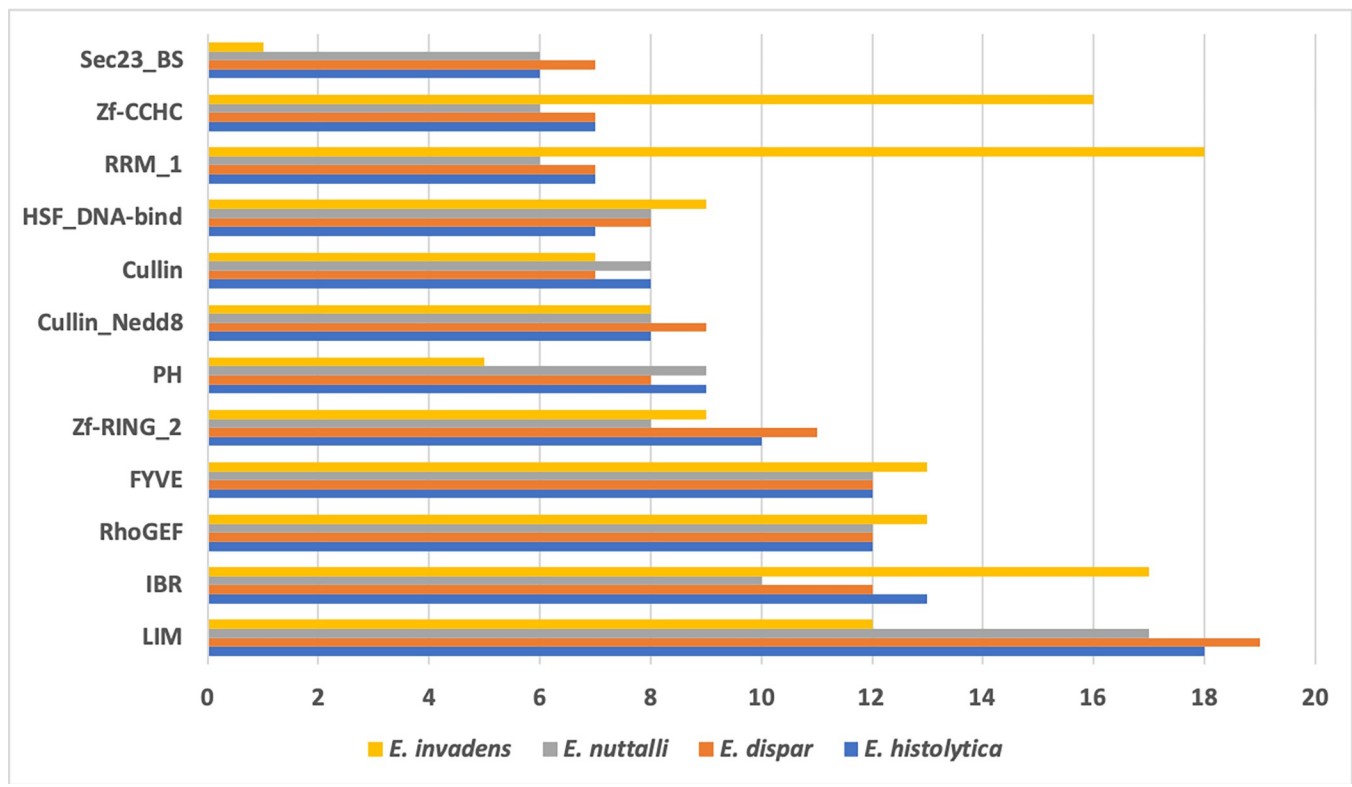

**Fig 3. The most abundant families in *Entamoeba* strains.** On the X-axis is the number of proteins; the Y-axis indicates the TF family names.

exclusive TFs. In the case of *E. histolytica* and *E. dispar*, the species are morphologically indistinguishable in both the cyst and the trophozoite forms [51]. Therefore, on the one hand, *E. dispar* must have a series of genes that are repressed or activated and prevent it from causing disease in its host [52], whereas *E. histolytica* seems to have a series of mechanisms, including transcriptional control, that allow it to cause amoebic colitis or a liver abscess in its host, or not. Some of these molecules are enzymes such as glycosidases (sialidase, N-acetylgalactosaminidase, and N-acetylglucosaminidase) which are necessary for the invasion of the epithelium, cysteine proteases (CP-A4, CP-A6, EhCPADH, and CP-B1, among others) which kill inflammatory and epithelial cells; or proteins such as the amoebopore that causes cell cytolysis, or the Gal/GalNAc lectin, which is involved in adhesion and the cytopathic effect [53,54] *E. nuttalli*, whose host is the macaque, has a life cycle similar to that of other *Entamoeba* species, and it is also capable of infecting humans but generates an asymptomatic infection. This suggests that this species must also present genomic plasticity and, therefore, specific transcription patterns that require a greater versatility of TFs. Therefore, a finely controlled transcriptional regulation must be carried out in which different TFs are necessary, hence the possible versatility of the families of TFs found in this work. However, we also observed that there are conserved TF families in the four species (Fig 3), in which the similar basic TF repertoire can be found, which may be performing basal transcription in these organisms. Finally, the species *E. histolytica*, *E. dispar*, and *E. nuttalli* present a similar number of TFs from the most abundant families, which coincides with the fact that the three species present a close phylogenetic relationship [51].

In general, few of the TFs identified in this work by sequence analysis have been previously characterized, such as, Nuclear factor Y (NF-Y) that appears at a later time point of *Entamoeba*

encystation [55–57]; the Ehp53, homologous to the tumor suppressor protein p53 [58] from human and *Drosophila melanogaster*. Ehp53 contains seven of the eight DNA-binding residues and two of the four Zn2+-binding sites described for p53. Heterologous monoclonal antibodies against p53 (Ab-1 and Ab-2) recognized a single 53 kDa spot in two-dimensional gels and they inhibited the formation of DNA-protein complexes produced by the interaction of nuclear extracts of *E. histolytica* with an oligonucleotide containing the consensus sequence for the binding of human p53. [58]. In addition, a calcium-sensitive EF-hand protein that binds to the URE3 motif [59], and two proteins that bind to the URE4 sequence (EhEBP1 and EhEBP2) [60], have also been identified.

Finally, we identified by sequence comparisons, members of the Myb-SHAQKYF family. These proteins have been previously identified as differentially expressed in trophozoites under basal cell culture conditions. Members of this group harbor a highly conserved and structured Myb-DBD and a large portion of intrinsically disordered residues. As the Myb-DBD of these proteins harbors a distinctive Q[VI]R[ST]HAQK[YF]F sequence in its putative third α-helix. An NMR structure of the Myb-DBD of EhMybS3 shows that this protein is composed of three α-helices stabilized by a hydrophobic core, similar to Myb proteins of different kingdoms [61]. Therefore, our approach opens the possibility to characterize experimentally diverse TFs with hypothetical functions, predicted in this work.

## Regulatory networks

The GRN is defined as a graph $G = (V, A)$, where $V$ is a set of nodes that correspond to genes or proteins in the network and $A$ is a set of edges that correspond to relationships between two nodes. Few GRNs have been reconstructed from experimental data; comparative genomics approaches are usually used for the reconstruction of GRNs in little-known organisms. To this end, the GRN from a model organism can be used as a template to export interactions in the organism of interest; under this approach, orthologous TFs generally regulate the expression of orthologous TGs [28,62]. Therefore, to identify the GRNs of the four *Entamoeba* species, the GRNs of *S. cerevisiae* and *H. sapiens* were used as references. A regulatory association was established when orthologues of a TF-TG relationship in a model organism were found for both a TF and a TG in the target organism (Table 1) (S2 Table) [28,62].

The inferred GRN of *E. histolytica* has 221 nodes and 272 interactions. The regulatory interactions conserved were preferentially assigned from *S. cerevisiae* (248 interactions), and also from the interactions from *H. sapiens* (24 interactions) (Fig 4A). The networks include 28 regulatory proteins: 18 were inferred by homology, and 10 were inferred based on InterPro and Pfam assignments. Of these 28, 4 TFs are self-regulated, *i.e.*, the TF regulates its own gene.

The GRN of *E. dispar* has 583 nodes and 979 interactions. The regulatory interactions conserved were preferentially assigned from *S. cerevisiae* (942 interactions), whereas 34 interactions were inferred from *H. sapiens* (Fig 4B). The networks include 41 regulatory proteins, 17 of which were inferred by homology and 24 were inferred by InterPro and Pfam assignments; from these 41, 7 TFs are self-regulated.

The GRN of *E. nuttalli* includes 382 nodes and 560 interactions. From these, 582 regulatory interactions were inferred from *S. cerevisiae* and 23 interactions from *H. sapiens* (Fig 4C). The network also includes 39 regulatory proteins, of which 6 are self-regulated.

Finally, the GRN of *E. invadens* has 520 nodes and 853 interactions. The regulatory interactions were preferentially assigned from *S. cerevisiae* (805 interactions) and also *H. sapiens* (50 interactions) (Fig 4D). The networks include 38 regulatory proteins, of which 7 TFs are self-regulated. Interestingly, the number of regulatory proteins in *E. histolytica* is lower and they appear to have fewer interactions and nodes compared to the regulatory proteins identified in

**Table 1. General properties of the GRNs.**

| GRN | *E. histolytica* | *E. dispar* | *E. nuttalli* | *E. invadens* |
|---|---|---|---|---|
| Total number of nodes | 221 | 583 | 382 | 520 |
| Total number of interactions | 272 | 979 | 560 | 853 |
| Number of TFs | 28 | 41 | 39 | 38 |
| Number of TGs | 205 | 542 | 521 | 482 |
| Self-regulated* | 4 | 7 | 6 | 7 |
| Maximum out degree* | EHI_044890— Helicase (121)** | EDI_297980—Hypothetical protein (284) | ENU1_120230—HSF (195) | EIN_249270—NF-Y alpha (257) |
| Maximum in-degree* | EHI_131470— NOP10 (4)*** | EDI_107330 –Xaa-Pro dipeptidase (7) | ENU1_214880 –Xaa-Pro dipeptidase (6) | EIN_095830—branched-chain-amino-acid aminotransferase (9) |
| Connected components | 4 | 5 | 3 | 5 |
| Average clustering | 0.0033 | 0.0723 | 0.043 | 0.0786 |

*This information can be found in S4 Table.

**The values in parentheses correspond to its Kin.

***The values in parentheses correspond to its Kout.

*E. dispar*, *E. nuttalli*, and *E. invadens*. For example, *E. dispar* has 13 more regulatory proteins than *E. histolytica*, but the number of nodes is more than double (521) and the number of interactions is three times higher than in *E. histolytica* (Table 1).

## Topological properties of the GRNs

In order to describe the global and local structures of the GRNs of *Entamoeba* spp. strains, the general structures of the four networks were analyzed (Table 1). Networks are structured into connected components (CC), within which the giant component is the one that contains the largest number of nodes in a network. In this context, we identified that the *E. histolytica* network comprises four CCs and the giant component contains 210 nodes and 263 edges; the *E. dispar* network contains five CCs and the giant component has 553 nodes and 945 edges; *E. nuttalli* contains three CCs and the giant component has 379 nodes and 558 edges. Finally, *E. invadens* contains five CCs and one giant component with 510 nodes and 846 edges (Fig 4).

In addition, the clustering coefficient, a measure of the degree to which nodes in a graph tend to cluster together, was calculated. We observed that the maximum clustering coefficient was 1 in *E. invadens* and 0.5 in the other networks. A clustering coefficient of 1 indicates that nodes with neighbors that are related between them form complete graphs, while a clustering coefficient less than 1 is related to few nodes being connected, which is common in the four networks due to limited information.

The input degree (*Kin*) and output degree (*Kout*), which link the number of TFs that control a gene and the number of genes that a TF regulates, were computed. In this context, we identified that 155 in *E. histolytica*, 295 in *E. dispar*, 234 in *E. nuttalli*, and 290 in *E. invadens* are regulated by one TF, *i.e.*, they have an input degree of 1. In this context, the most regulated gene in *E. histolytica* is EHI_131470 (ribosome biogenesis protein Nop10) responsible for ribosome biogenesis and is regulated by four TFs. EDI_107330 (Xaa-Pro dipeptidase) is regulated by seven TFs in *E. dispar*; in *E. nuttalli* ENU1_214880 (Xaa-Pro dipeptidase) is regulated by six TFs. Finally, in *E. invadens* EIN_095830 (branched-chain-amino-acid aminotransferase) is regulated by nine TFs (Table 1).

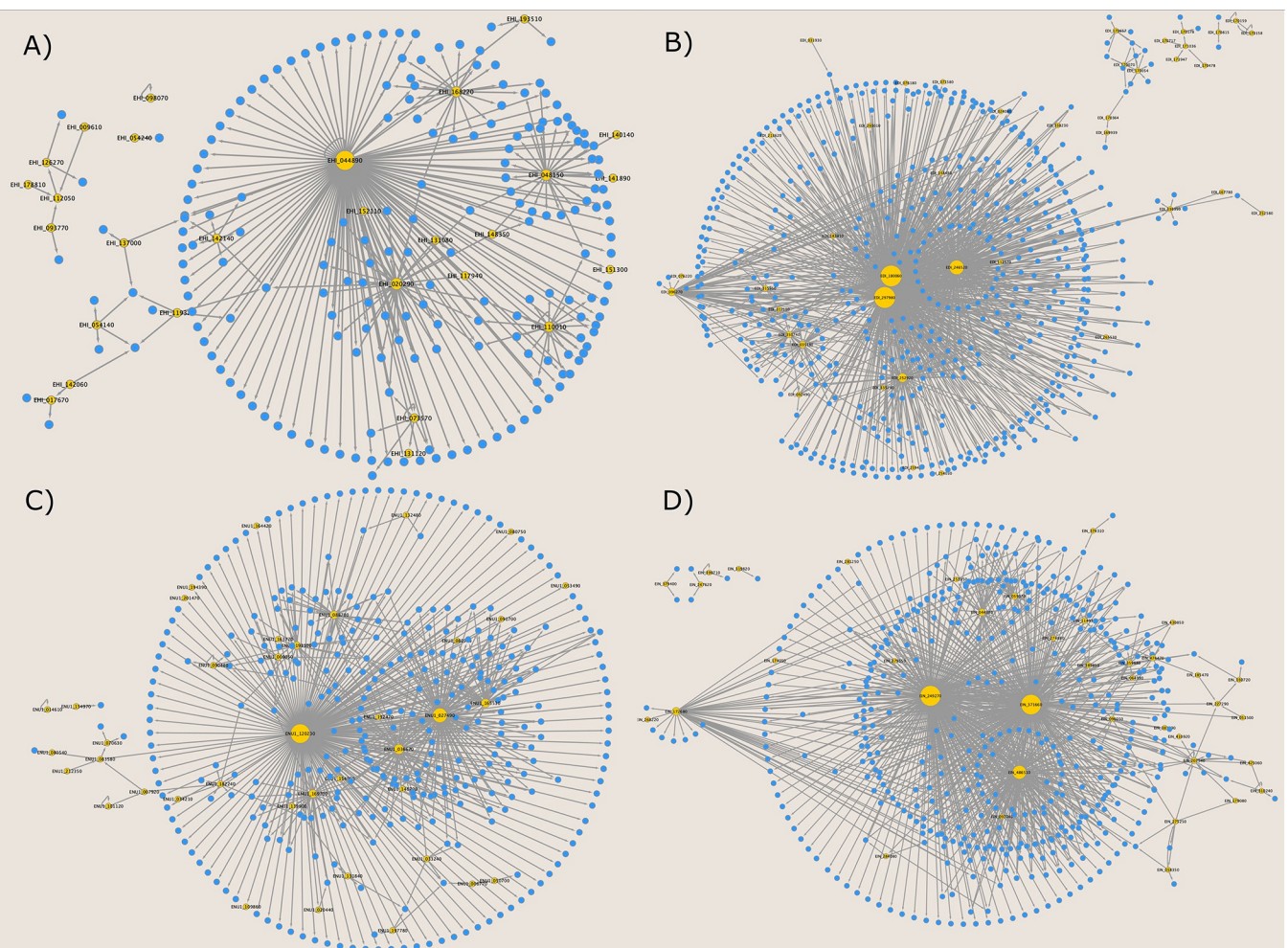

**Fig 4.** GRNs of A) *E. histolytica*, B) *E. dispar*, C) *E. nuttalli*, and D) *E. invadens*. The yellow nodes are TFs and the blue nodes are TGs; size nodes are proportional to output degree.

With regard to output degree, the most connected node in the *E. histolytica* network is the putative Helicase EHI_044890, which influences 121 genes. It is interesting that the orthologous proteins of EHI_044890 in other organisms are essential, according to the TDR targets database (https://www.tdrtargets.org). For instance, in *Trypanosoma brucei*, its mutation reduces significant loss of fitness in differentiation of procyclic to bloodstream forms, whereas in *Caenorhabditis elegans*, it is lethal for the embryonic stage. Therefore, we suggest that EHI_044890 is also essential in *E. histolytica*, because of its output degree, functional role, and similarity to other proteins.

EDI_297980 regulates 284 genes and is a hypothetical protein in *E. dispar* network. ENU1_120230 regulates 195 genes and is a putative heat shock transcription factor in *E. nuttalli*. This type of TFs has not been identified in any *Entamoeba* species, except in *E. histolytica*, which has a family of seven EhHSTFs (manuscript in preparation). EhHSTF7 has recently been shown to be the TF responsible for regulating the expression of the multidrug resistance gene EhPgp5 in this species of amoeba [57].

Finally, EIN_249270 regulates 257 genes and is a putative transcription factor NF-Y alpha in *E. invadens*. This TF has been a heterotrimeric protein composed of NF-YA, NF-YB, and NF-YC subunits that bind to the CCAAT box. This factor participates mainly in the regulation

of genes of the cell cycle and metabolism such as gluconeogenesis and appears at a later time point of *Entamoeba* encystation [55–57,63]. In addition, we identified the top 10 most important nodes by the centrality metrics, based on node connectivity as well as the shortest paths between them. In terms of degree centrality, the most important nodes in each network included EHI_044890 (0.5545), which is an isw2p helicase in *E. histolytica* [64]; its homologues in *S. cerevisiae* are chromatin-remodeling factors and yeast ISWI, which is essential for the cell to resist various stresses in vivo, and both homologous show genetic interactions [65]. EDI_297980 (0.4914) is a hypothetical protein in *E. dispar*, and it is homologous to nuclear transcription factor Y, alpha (KEGG). In *E. nuttalli*, ENU1_120230 (0.5118) is orthologous with YGL073W, which is a trimeric heat shock TF [66]. Finally, EIN_249270 (0.4971), which is orthologous to nuclear transcription factor Y, alpha (KEGG), is the most important in *E. invadens*.

Furthermore, we identified the node with the highest closeness score, *i.e.*, that which minimizes the sum of distances to the other nodes. In *E. histolytica* the most important is EHI_131470 (0.0189), for ribose biogenesis protein Nop10, involved in 18S rRNA pseudouridylation and in cleavage of pre-rRNA [67]. In *E. dispar*, EDI_107330 (0.0126) is a putative Xaa-Pro dipeptidase; Xaa-Pro dipeptidase plays a role in collagen metabolism because of the high level of imino acids in collagen (Uniprot). In *E. nuttalli*, ENU1_214880 (0.0167) is also homologous with Xaa-Pro dipeptidase. In *E. invadens* EIN_095830 (0.0166) is a putative branched-chain amino acid aminotransferase.

Betweenness centrality of a node is defined as the sum of the fraction of all-pairs shortest paths that pass through *v*, *i.e.*, the influence of a vertex over the flow of information between every pair of vertices under the assumption that information primarily flows over the shortest paths between them. The most important in *E. histolytica* is EHI_048150 (0.0011519), which encodes the EhCAF1 protein homologous to POP2 in *S. cerevisiae*; POP2 is a nuclease involved in mRNA deadenylation [68,69]. EDI_297980 (0.000999) in *E. dispar* and EIN_249270 (0.0011270) in *E. invadens* are orthologs to nuclear transcription factor Y, alpha. ENU1_027490 (0.00050421) in *E. nuttalli* is homologous to ISW2, a conserved ATP-dependent chromatin-remodeling factor in *S. cerevisiae* [70].

## Biological process in the networks

To identify the most abundant functions represented in the networks, they were analyzed with Gene Ontology terms. We identified that the most abundant terms in the four networks are Macromolecule metabolic process (GO:0043170), Cellular macromolecule metabolic process (GO:0044260), and Cellular nitrogen compound metabolic process (GO:0034641), all of which are related to chemical reactions and pathways involving macromolecules and organic and inorganic nitrogenous compounds (Fig 5) (S3 Table).

Additionally, we identified some GO terms associated with one species' network: single-organism biosynthetic process (GO:0044711), organophosphate metabolic process (GO:0019637), and macromolecular complex subunit organization (GO:0043933) in *E. histolytica*; single-organism carbohydrate metabolic process (GO:0005975) and organic substance catabolic process (GO:1901575) in *E. dispar*; generation of precursor metabolites and energy (GO:0006091) in *E. nuttalli*; regulation of protein complex assembly (GO:0043254), regulation of actin filament-based process (GO:0032970), and generation of precursor metabolites and energy (GO:0006091) in *E. invadens*.

## Conclusions

The inference of GRNs of *Entamoeba* speciesprovide an excellent opportunity to understand how genes and functional processes are interrelated in these organisms. These networks were

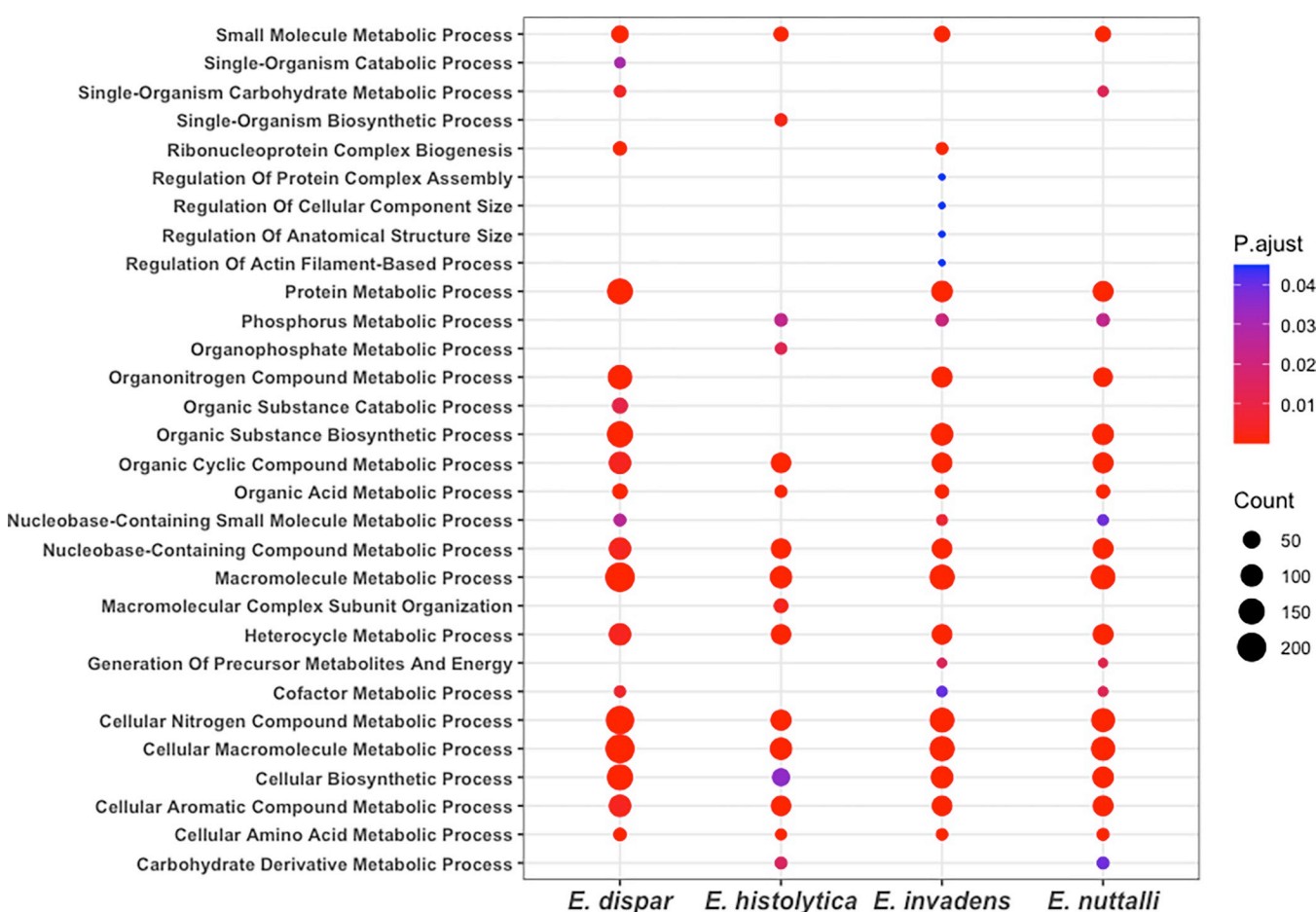

**Fig 5. GO enrichment analysis.** The dot plot shows the terms (FDR < 0.05) of biological processes identified using DAVID. The size of a dot represents the number of genes associated with the GO term, and the color of dots represents the P-adjusted value.

analyzed in terms of these topological properties to infer the role of TFs in the context of the GRN and the biological functions. From these analyses, we identified that TFs of *E. histolytica*, *E. dispar*, and *E. nuttalli* are associated with the LIM family, whereas the TFs in *E. invadens* are associated with the RRM_1 family. In the context of more connected nodes, we identified that EHI_044890 regulates 121 genes in *E. histolytica*, EDI_297980 regulates 284 genes in *E. dispar*, ENU1_120230 regulates 195 genes in *E. nuttalli*, and EIN_249270 regulates 257 genes in *E. invadens*. Finally, we determined that Macromolecule metabolic process (GO:0043170), Cellular macromolecule metabolic process (GO:0044260), and Cellular nitrogen compound metabolic process (GO:0034641) are the main biological processes for each network. However, there are specific enriched biological processes for each network that determine the differences in the size of each network. The results described in this work can be used for the study of gene regulation in these organisms.

## Supporting information

**S1 Table. Transcription factors of *Entamoebas* strains.**
(XLSX)

**S2 Table. Gene regulatory networks of *Entamoebas* strains.**
(XLSX)

**S3 Table. Gene ontology terms of *Entamoebas* networks.**
(XLSX)

**S4 Table. Main topological measures.**
(XLSX)

**S1 Data. Script to build a network from a template.**
(ZIP)

## Acknowledgments

We thank Israel Sanchez, Manuel Lira, and Suyin Ortega for their technical support.

## Author Contributions

**Conceptualization:** Edgardo Galán-Vásquez, Ernesto Pérez-Rueda.

**Data curation:** Edgardo Galán-Vásquez.

**Formal analysis:** Edgardo Galán-Vásquez, Ernesto Pérez-Rueda.

**Funding acquisition:** Edgardo Galán-Vásquez, Ernesto Pérez-Rueda.

**Investigation:** María del Consuelo Gómez-García, Ernesto Pérez-Rueda.

**Methodology:** Edgardo Galán-Vásquez, María del Consuelo Gómez-García, Ernesto Pérez-Rueda.

**Supervision:** Ernesto Pérez-Rueda.

**Validation:** María del Consuelo Gómez-García.

**Writing – original draft:** Edgardo Galán-Vásquez, María del Consuelo Gómez-García, Ernesto Pérez-Rueda.

**Writing – review & editing:** Edgardo Galán-Vásquez, María del Consuelo Gómez-García, Ernesto Pérez-Rueda.

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
