## [Decision Letter · Decision Letter 0]

25 May 2022

PONE-D-22-06056A landscape of gene regulation in the parasitic amoebozoa Entamoeba spp.PLOS ONE

Dear Dr. Galán-Vásquez,

Thank you for submitting your manuscript to PLOS ONE. After careful consideration, we feel that it has merit but does not fully meet PLOS ONE’s publication criteria as it currently stands. Therefore, we invite you to submit a revised version of the manuscript that addresses the points raised during the review process.

We look forward to receiving your revised manuscript.

Kind regards,

Jesús Valdés, Ph.D.

Academic Editor

PLOS ONE

Journal Requirements:

“This work was supported by the Dirección General de Asuntos del Personal Académico-Universidad Nacional Autónoma de México (IA201221 and IN-209620). This work was supported by the Secretaría de Investigación y Posgrado, Instituto Politécnico Nacional [Grant SIP20211325].”

“This work was supported by the Dirección General de Asuntos del Personal Académico-Universidad Nacional Autónoma de México (IA201221 and IN-209620). This work was supported by the Secretaría de Investigación y Posgrado, Instituto Politécnico Nacional [Grant SIP20211325]. The funders had no role in study design, data collection and analysis, decision to publish, or preparation of the manuscript.”

- https://www.frontiersin.org/articles/10.3389/fimmu.2017.01237/full

- https://pubmed.ncbi.nlm.nih.gov/33730052/

In your revision ensure you cite all your sources (including your own works), and quote or rephrase any duplicated text outside the methods section. Further consideration is dependent on these concerns being addressed.

Additional Editor Comments :

Please submit a revised version of the manuscript addressing the concerns raised by one of the reviewers.

Reviewers' comments:

Reviewer's Responses to Questions

**Comments to the Author**

1. Is the manuscript technically sound, and do the data support the conclusions?

Reviewer #1: Partly

Reviewer #2: Yes

2. Has the statistical analysis been performed appropriately and rigorously? 

Reviewer #1: Yes

Reviewer #2: Yes

3. Have the authors made all data underlying the findings in their manuscript fully available?

Reviewer #1: No

Reviewer #2: Yes

4. Is the manuscript presented in an intelligible fashion and written in standard English?

Reviewer #1: No

Reviewer #2: No

5. Review Comments to the Author

Reviewer #1: In this work are presented bioinformatics data proposing gene regulatory networks (GRN) in four species of Entamoeba: E. histolytica, E. dispar, E. nuttalli and E. invadens. Transcription factors and the potential genes they target have been identified. Next, GRNs were attempted using information from GRNs from yeast and humans. The objective of this work is interesting and important to understand certain homologies and/or differences concerning gene regulation between these species, two of which infect humans (E. histolytica, E. dispar).

As presented, the data, and the manuscript, have important caveats that deserve special attention when considering an eventual publication in PLOS ONE.

1. The manuscrit mainly reports mathematical-computer approaches, which requires a pedagogical and understandable presentation for most readers of a generalist journal like PLOS ONE; even for a for a public familiar with genomics who may be interested in genetic regulation, as the tests proposed here are not described. The first request is the presentation of a workflow of the simulations and tests carried out. In the results chapter, legend of figures and tables, all data must be presented in full not only interpretations.

2. The analysis is based on findings of orthologous proteins between the four species of entamoeba and concludes with a test of enrichment genes when searching for GRN. Entamoeba species are known to exhibit many gene/protein families (e.g., BspA, AGI families). In these results, the contribution to these families is not considered although this may lead to false enrichment data. Is it the case in these results? ProteinOrtho and OrthoVenn2 were used: In which case? give them the same results? All these points need precision

3. Several Transcription Factors were identified, and this set is a good contribution. It should be interesting to present them in a clear way as the tables with only gene ID entries does not explain enough. In particular, the discussion on the only one, named transcription factor NF-Y, is wrong as this factor has been already described (PMID: 32766170; PMID:192394). Also, others transcription factors are known in Entamoeba; for instance, papers from Esther Orozco group, as example among many papers, see PMID: 29514716; or Carol Gilchrist PMID: 20689746 group and others as PMID: 30375973 are not cited and discussed.

4. Line 143. The main functions were determined with which set of genes.

5. The author said: "to identify the GRNs of the four Entamoeba species, the GRNs of S. cerevisiae and H. sapiens were used as references. A regulatory association was established when orthologues of a TF-TG relationship in a model organism were found for both a TF and a TG in the target organism" How it has been done, by eye inspection or computing? Because this experiment gave the main result presented in the paper; it deserves a detailed method. table 1 or Supplemental Table 4 does not correspond to the result but to the interpretation of these.

6. Important assumptions on Entamoeba biology are wrong:

-Line 52. parasitic and nonpathogenic species have been described, including Entamoeba histolytica. All Entamoeba are parasites! And E. histolytica is pathogenic

-Line 155 - 158. Entamoeba dispar also colonizes the human intestine, even it is the. most spread worldwide entamoeba species

-Lines 328 to 325. Huge mistake, E. nuttalli is a specie isolated from macaques and not for reptiles!

-Line 382. more import nodes....means?

7. Tables do not have a title, these tables also need precision for TF names and gene product description

Reviewer #2: It is a well written MS that deals with how genes and proteins interrelate to accomplish functional processes in cells. The authors study these interrelationships in four different species of Entamoebas: E. histolytica. E. dispar, E. nuttalli and E. invadens. The first two are found in humans, the third in monkeys, but can invade humans, and the last in reptiles. The authors use data programs to analyze genes and proteins and find that they tend to be in groups of genes and proteins that have to do with macromolecular metabolic processes, cellular macromolecular metabolic processes and cellular metabolic processes in which nitrogen participates. They observe that there are groups of genes that regulate a protein or a gene that is capable of regulating the functioning of several proteins. They also observe that there are differences between the total number of genes and proteins that make up each amoebic species and speculate that this could be due to the different habitats where they are found. Finally, they propose that all the work shown by the researchers can be used by other research groups to carry out more specific investigations of the genes and proteins that participate in these processes.

I think that it is an interesting piece of research and that it provides data in the literature on amoebic species, which should be used by other research groups.

6. PLOS authors have the option to publish the peer review history of their article (what does this mean?). If published, this will include your full peer review and any attached files.

Reviewer #1: No

Reviewer #2: No

---

## [Author Response · Author response to Decision Letter 0]

17 Jun 2022

Reviewer #1: In this work are presented bioinformatics data proposing gene regulatory networks (GRN) in four species of Entamoeba: E. histolytica, E. dispar, E. nuttalli and E. invadens. Transcription factors and the potential genes they target have been identified. Next, GRNs were attempted using information from GRNs from yeast and humans. The objective of this work is interesting and important to understand certain homologies and/or differences concerning gene regulation between these species, two of which infect humans (E. histolytica, E. dispar).

As presented, the data, and the manuscript, have important caveats that deserve special attention when considering an eventual publication in PLOS ONE.

1. The manuscrit mainly reports mathematical-computer approaches, which requires a pedagogical and understandable presentation for most readers of a generalist journal like PLOS ONE; even for a for a public familiar with genomics who may be interested in genetic regulation, as the tests proposed here are not described. The first request is the presentation of a workflow of the simulations and tests carried out. In the results chapter, legend of figures and tables, all data must be presented in full not only interpretations.

RESPONSE: We have included the following workflow as figure 1, describing the main steps of the inference and analysis of the GRN in E. histolytica.

Figure 1. Schematic workflow of the network inference procedure steps. Four Entamoeba genomes were downloaded from NCBI and compared with Orthovenn, to infer the common set of orthologous proteins. Interproscan was used to assign protein domains (PFAM and Supfam). To infer the GRN, two organismal models were used, S.cerevisiae and H. sapiens, and the ProteinOrtho was considered. The GRNs inferred were evaluated in terms of their topological properties, hubs, and functional descriptions.

2. The analysis is based on findings of orthologous proteins between the four species of entamoeba and concludes with a test of enrichment genes when searching for GRN. Entamoeba species are known to exhibit many gene/protein families (e.g., BspA, AGI families). In these results, the contribution to these families is not considered although this may lead to false enrichment data. Is it the case in these results? ProteinOrtho and OrthoVenn2 were used: In which case? give them the same results? All these points need precision

RESPONSE: We agree with the reviewer. We have clarified the difference of proteinortho and orthovenn in the manuscript. In this regard, proteinortho was used to infer the TF - TG interactions from two models (S. cerevisiae and H. sapiens) where the GRNs have been characterized. In contrast, OrthoVenn2 was used to identify common orthologs between the four Entamoeba species and to infer the most significant functions. We have modified the sections in methodology to clarify the assignments, and inserted the follow paragraphs (highlighted in the main manuscript):

“To identify orthologous proteins between the four Entamoeba spp. proteomes and the proteomes of S. cerevisiae and H. sapiens, were used to identify the orthologs using we used the program ProteinOrtho (V5.16) [36], with the following parameters: E-value of 0.01, a sequence coverage ≥ of 50%, and minimal percent identity of best Blast hits of 30%.

To infer the GRNs, we map their interactions considering the following criteria: If the orthologs of the TF and its TG of the model organism (S. cerevisiae and / or H. sapiens,) were found in a new genome, the interaction was assigned using guilt by association. Then, each network was integrated using all the ortholog assignments with the two six reference GRNs. All the network interactions can be inferred by running the scripts, provided as supplementary data 1.”

Concerning the role of families in the inference of the GRN, they were not considered to identify the TF - TG interactions. To this end, we used a guilt-by-association approach; i.e., if the TF - TG of S. cerevisiae is identified by orthology in Entamoeba, the interaction is assigned, under the hypothesis that the pair of orthologs are functionally related as the reference genome. Therefore, the family definition is not used, because multiple evolutionary events can be associated to them, beyond orthology, as paralogy or xenology; however we could consider in further analysis.

3. Several Transcription Factors were identified, and this set is a good contribution. It should be interesting to present them in a clear way as the tables with only gene ID entries does not explain enough. In particular, the discussion on the only one, named transcription factor NF-Y, is wrong as this factor has been already described (PMID: 32766170; PMID:192394). Also, others transcription factors are known in Entamoeba; for instance, papers from Esther Orozco group, as example among many papers, see PMID: 29514716; or Carol Gilchrist PMID: 20689746 group and others as PMID: 30375973 are not cited and discussed.

RESPONSE: We have included additional discussion about TFs identified in this work and described experimentally in previous works. In addition, we have corrected the information concerning the transcription factor NF-Y.

We have inserted the following paragraphs at the end of the TF prediction section.

“In general, few of the TFs identified in this work by sequence analysis have been previously characterized, such as, Nuclear factor Y (NF-Y) that appears at a later time point of Entamoeba encystation [55-57]; the Ehp53, homologous to the tumor suppressor protein p53 [58] from human and Drosophila melanogaster. Ehp53 contains seven of the eight DNA-binding residues and two of the four Zn2+-binding sites described for p53. Heterologous monoclonal antibodies against p53 (Ab-1 and Ab-2) recognized a single 53 kDa spot in two-dimensional gels and they inhibited the formation of DNA-protein complexes produced by the interaction of nuclear extracts of E. histolytica with an oligonucleotide containing the consensus sequence for the binding of human p53. [58]. In addition, a calcium-sensitive EF-hand protein that binds to the URE3 motif [59], and two proteins that bind to the URE4 sequence (EhEBP1 and EhEBP2) [60], have also been identified.

Finally, we identified by sequence comparisons, members of the Myb-SHAQKYF family. These proteins have been previously identified as differentially expressed in trophozoites under basal cell culture conditions. Members of this group harbor a highly conserved and structured Myb-DBD and a large portion of intrinsically disordered residues. As the Myb-DBD of these proteins harbors a distinctive Q[VI]R[ST]HAQK[YF]F sequence in its putative third α-helix. An NMR structure of the Myb-DBD of EhMybS3 shows that this protein is composed of three α-helices stabilized by a hydrophobic core, similar to Myb proteins of different kingdoms [61]. Therefore, our approach opens the possibility to characterize experimentally diverse TFs with hypothetical functions, predicted in this work.”

55.- Manna D, Lentz CS, Ehrenkaufer GM, Suresh S, Bhat A, Singh U. An NAD+-dependent novel transcription factor controls stage conversion in entamoeba. Elife. 2018; 7:e37912. 10.7554/eLife.37912.030.

56.- Manna D, Singh U. (2019). Nuclear Factor Y (NF-Y) Modulates Encystation in Entamoeba via Stage-Specific Expression of the NF-YB and NF-YC Subunits. mBio. 2019; 10(3), e00737-19. https://doi.org/10.1128/mBio.00737-19.

57.- Bello F, Orozco E, Benítez-Cardoza CG, Zamorano-Carrillo A, Reyes-López CA, Pérez-Ishiwara DG, et al. The novel EhHSTF7 transcription factor displays an oligomer state and recognizes a heat shock element in the Entamoeba histolytica parasite. Microb Pathog 2022; 162;105349. doi: 10.1016/j.micpath.2021.105349.

58.- Mendoza L, Orozco E, Rodríguez MA, García-Rivera G, Sánchez T, García E, et al. Ehp53, an Entamoeba histolytica protein, ancestor of the mammalian tumour suppressor p53. Microbiology (Reading, England). 2003; 149(Pt 4), 885–893. https://doi.org/10.1099/mic.0.25892-0

59.- Gilchrist CA, Holm CF, Hughes MA, Schaenman JM, Mann BJ, Petri Jr WA. Identification and characterization of an Entamoeba histolytica upstream regulatory element 3 sequence-specific DNA-binding protein containing EF-hand motifs. J. Biol. Chem. 2001; 276, pp. 11838-11843.

60.- Schaenman JM, Gilchrist CA, Mann BJ, Petri Jr WA. Identification of two Entamoeba histolytica sequence-specific URE4 enhancer-binding proteins with homology to the RNA-binding motif RRM. J. Biol. Chem. 2001; 276, pp. 1602-1609.

61.- Cárdenas-Hernández H, Titaux-Delgado GA, Castañeda-Ortiz EJ, Torres-Larios A, Brieba LG, del Río-Portilla F, et al. Genome-wide and structural analysis of the Myb-SHAQKYF family in Entamoeba histolytica. Biochimica et Biophysica Acta (BBA)-Proteins and Proteomics. 2021; 1869(4), 140601.

4. Line 143. The main functions were determined with which set of genes.

RESPONSE: We have clarify this sentence:

“To identify the most abundant functions represented in the networks, they were analyzed with Gene Ontology terms.”

5. The author said: "to identify the GRNs of the four Entamoeba species, the GRNs of S. cerevisiae and H. sapiens were used as references. A regulatory association was established when orthologues of a TF-TG relationship in a model organism were found for both a TF and a TG in the target organism" How it has been done, by eye inspection or computing? Because this experiment gave the main result presented in the paper; it deserves a detailed method. table 1 or Supplemental Table 4 does not correspond to the result but to the interpretation of these.

RESPONSE: We have expanded this explanation in material and methods. See answer to question 2.

6. Important assumptions on Entamoeba biology are wrong:

-Line 52. parasitic and nonpathogenic species have been described, including Entamoeba histolytica. All Entamoeba are parasites! And E. histolytica is pathogenic

RESPONSE: We have modified this sentence and included the corresponding references: 

5.- Espinosa A, Paz-y-Miño CG. Discrimination experiments in Entamoeba and evidence from other protists suggest pathogenic amebas cooperate with kin to colonize hosts and deter rivals. Journal of Eukaryotic Microbiology. 2019; 66(2), 354-368.

7.- Babuta M, Bhattacharya S, Bhattacharya A. Entamoeba histolytica and pathogenesis: A calcium connection. PLoS Pathogens. 2020; 16(5), e1008214.

8.- König C, Honecker B, Wilson IW, Weedall GD, Hall N, Roeder T, Metwally NG, Bruchhaus I. Taxon-Specific Proteins of the Pathogenic Entamoeba Species E. histolytica and E. nuttalli. Front. Cell. Infect. Microbiol. 2021; 11:641472. doi: 10.3389/fcimb.2021.641472.

-Line 155 - 158. Entamoeba dispar also colonizes the human intestine, even it is the. most spread worldwide entamoeba species

RESPONSE: We have corrected this mistake and modified the paragraph:

“Finally, Entamoeba dispar is considered a nonpathogenic species that lives as a commensal in humans; first described in 1993 for Diamond and Clark [14]. was posteriorly identified as able to produce liver and intestinal lesions that were occasionally indistinguishable from those produced by E. histolytica [15].”

-Lines 328 to 325. Huge mistake, E. nuttalli is a specie isolated from macaques and not for reptiles!

RESPONSE: Sorry for this mistake. We have corrected the isolation source to “macaques” and added the adequate references.

11.- Tachibana H, Yanagi T, Pandey K, Cheng XJ, Kobayashi S, Sherchand JB, et al. An Entamoeba sp. strain isolated from rhesus monkey is virulent but genetically different from Entamoeba histolytica. Molecular and biochemical parasitology. 2007; 153(2), 107-114.

12.- Tachibana H, Yanagi T, Lama C, Pandey K, Feng M, Kobayashi S, et al. Prevalence of Entamoeba nuttalli infection in wild rhesus macaques in Nepal and characterization of the parasite isolates. Parasitology International. 2013; 62(2), 230-235.

-Line 382. more import nodes....means?

RESPONSE: We have modified the sentence to:

“In the context of more connected nodes…”

7. Tables do not have a title, these tables also need precision for TF names and gene product description

RESPONSE: We have included titles and more information in the tables.

 

Reviewer #2: It is a well written MS that deals with how genes and proteins interrelate to accomplish functional processes in cells. The authors study these interrelationships in four different species of Entamoebas: E. histolytica. E. dispar, E. nuttalli and E. invadens. The first two are found in humans, the third in monkeys, but can invade humans, and the last in reptiles. The authors use data programs to analyze genes and proteins and find that they tend to be in groups of genes and proteins that have to do with macromolecular metabolic processes, cellular macromolecular metabolic processes and cellular metabolic processes in which nitrogen participates. They observe that there are groups of genes that regulate a protein or a gene that is capable of regulating the functioning of several proteins. They also observe that there are differences between the total number of genes and proteins that make up each amoebic species and speculate that this could be due to the different habitats where they are found. Finally, they propose that all the work shown by the researchers can be used by other research groups to carry out more specific investigations of the genes and proteins that participate in these processes.

I think that it is an interesting piece of research and that it provides data in the literature on amoebic species, which should be used by other research groups.

RESPONSE: We appreciate the time to read the manuscript and make the positive comment.

---

## [Editor Report · Decision Letter 1]

6 Jul 2022

A landscape of gene regulation in the parasitic amoebozoa Entamoeba spp.

PONE-D-22-06056R1

Dear Dr. Galán-Vásquez,

We’re pleased to inform you that your manuscript has been judged scientifically suitable for publication and will be formally accepted for publication once it meets all outstanding technical requirements.

Kind regards,

Jesús Valdés, Ph.D.

Academic Editor

PLOS ONE
---

## [Editor Report · Acceptance letter]

21 Jul 2022

PONE-D-22-06056R1 

A landscape of gene regulation in the parasitic amoebozoa Entamoeba spp. 

Dear Dr. Galán-Vásquez:

I'm pleased to inform you that your manuscript has been deemed suitable for publication in PLOS ONE. Congratulations! Your manuscript is now with our production department. 

Kind regards, 

on behalf of

Dr. Jesús Valdés 

Academic Editor

PLOS ONE